# Development of New Diffuse Large B Cell Lymphoma Mouse Models

**DOI:** 10.3390/cancers16173006

**Published:** 2024-08-29

**Authors:** Syed Hassan Mehdi, Ying-Zhi Xu, Leonard D. Shultz, Eunkyung Kim, Yong Gu Lee, Samantha Kendrick, Donghoon Yoon

**Affiliations:** 1Myeloma Center, University of Arkansas for Medical Sciences, Little Rock, AR 72205, USA; shmehdi@uams.edu (S.H.M.); ekim@uams.edu (E.K.); 2Department of Biochemistry and Molecular Biology, University of Arkansas for Medical Sciences, Little Rock, AR 72205, USA; yxu2@uams.edu; 3The Jackson Laboratory, Bar Harbor, ME 04609, USA; lenny.shultz@jax.org; 4College of Pharmacy, Institute of Pharmaceutical Sciences and Technology, Hanyang University, Ansan 15588, Republic of Korea; yglee1721@hanyang.ac.kr

**Keywords:** diffuse large B cell lymphoma, humanized mouse model, interleukin-6 (IL6), germinal center B cell-like (GCB), activated B cell-like (ABC)

## Abstract

**Simple Summary:**

Diffuse large B cell lymphoma (DLBCL) is the most diagnosed, aggressive non-Hodgkin lymphoma, a type of blood cancer. In this study, we developed new DLBCL cell-derived xenograft mouse models. We found that our models show consistent tumor burden with unformal disease progression and organ-specific infiltration.

**Abstract:**

Diffuse large B cell lymphoma (DLBCL) is the most diagnosed, aggressive non-Hodgkin lymphoma, with ~40% of patients experiencing refractory or relapsed disease. Given the low response rates to current therapy, alternative treatment strategies are necessary to improve patient outcomes. Here, we sought to develop an easily accessible new xenograft mouse model that better recapitulates the human disease for preclinical studies. We generated two Luciferase (Luc)-EGFP-expressing human DLBCL cell lines representing the different DLBCL cell-of-origin subtypes. After intravenous injection of these cells into humanized NSG mice, we monitored the tumor growth and evaluated the organ-specific engraftment/progression period. Our results showed that human IL6-expressing NSG (NSG-IL6) mice were highly permissive for DLBCL cell growth. In NSG-IL6 mice, systemic engraftments of both U2932 activated B cell-like- and VAL germinal B cell-like-DLBCL (engraftment rate; 75% and 82%, respectively) were detected within 2nd-week post-injection. In the organ-specific ex vivo evaluation, both U2932-*Luc* and VAL-*Luc* cells were initially engrafted and expanded in the spleen, liver, and lung and subsequently in the skeleton, ovary, and brain. Consistent with the dual *BCL2*/*MYC* translocation association with poor patient outcomes, VAL cells showed heightened proliferation in human IL6-conditioned media and caused rapid tumor expansion and early death in the engrafted mice. We concluded that the U2932 and VAL cell-derived human IL6-expressing mouse models reproduced the clinical features of an aggressive DLBCL with a highly consistent pattern of tumor development. Based on these findings, NSG mice expressing human IL6 have the potential to serve as a new tool to develop DLBCL xenograft models to overcome the limitations of standard subcutaneous DLBCL xenografts.

## 1. Introduction

Diffuse large B cell lymphoma (DLBCL) is the most common subtype of non-Hodgkin lymphoma (NHL), constituting 25% to 30% of cases, with approximately 150,000 patients diagnosed annually worldwide [1]. DLBCL is a genetically complex hematological malignancy with two major subtypes, the activated B cell-like (ABC) and the germinal center B cell-like (GCB), that correlate with patient prognosis [2]. Although much progress has been made in treating this disease in that 60% of DLBCL patients will achieve long-term remission with contemporary immuno-chemotherapy [3], the remaining 40% experience primary refractory disease or relapse [4].

Several spontaneous and induced B cell lymphoma animal models have been developed using transgenic mice or transplanting with various types of tumor cells to understand mechanisms underlying DLBCL progression [5,6,7,8,9]. These transgenic or transplant xenograft models are limited, as they often fail to recapitulate the heterogeneous subclassifications of this complex disease. While transgenic immunocompetent mice allow for spontaneous tumor formation, these models rely on inducing the expression of specific oncogenes that drive a select group of DLBCL. The transplant xenograft model offers several advantages, such as reproducing late-stage disease and shortening the time of disease development. Despite these benefits, the current xenograft models are also limited by variable reproducibility and an inability to assess interactions with tumor micro-environments. Recent studies found that interleukin-6 (IL6) is an important growth factor for normal and malignant B cell engraftment and growth [10,11,12,13,14]. The IL6 protein sequence has a 42% similarity between mice and humans. While human IL6 works in both human and mouse cell assays, mouse IL6 does not elicit a response in human cells [15]. The human IL6 presence in the mouse facilitates the engraftment of malignant cells, although no engraftment rate was reported [10,14]. Hashwah et al. reported that genetically humanized mouse strains, the MISTRG and MISTRG6, supported better growth of DLBCL cell lines and primary DLBCL cells with inconsistent organ-infiltration patterns via orthotopic injection [14,16]. The MISTRG mouse strain expresses human SIRPα, M-CSF, IL-3, GM-CSF, and Thrombopoietin on the immune-deficient mouse (Rag2^−/−^/IL2Rγ^−/−^ background) under the control of mouse regulatory elements that provide increased support for the development and function of human monocytes, macrophages, and NK cells [17]. The MISTRG6 mouse is a modified MISTRG with an additional knock-in human IL6 allele and has previously been shown to improve multiple myeloma engraftments and growth [10].

Here, we employed two different subtypes of DLBCL cells, U2932 (ABC) and VAL (GCB), and intravenously injected them into three strains of immunodeficient mice, NOD-scid-IL2rgnull (NSG) mice expressing human IL6 (NSG-IL6), IL3/CSF2/KITLG (NSG-SGM3), or NSG mice expressing all four human cytokines (NSG-IL6/SGM3), to establish a pipeline for the rapid and reliable generation of in vivo DLBCL models. We evaluated these cell-derived xenograft models for early detection, the disease progression pattern, and the organ-specific engraftment and growth.

## 2. Materials and Methods

### 2.1. Mouse Handling and Maintenance

All mouse strains were housed and bred at the University of Arkansas for Medical Sciences (UAMS) Animal Facility. All animal-handling procedures were reviewed and approved by the UAMS IACUC (AUP #3987 and IPROTO202200000477) and were conducted as per the National Institutes of Health (NIH) Guide for the Care and Use of Laboratory Animals. Mice were kept on a 12 h light/12 h dark cycle. Room temperature was maintained at 24–26 °C. Food pellets and water were sterilized and provided as much as necessary or desired.

### 2.2. Mouse Strains and Genotyping

NOD.Cg-*Prkdc^scid^ Il2rg^tm1Wjl^* Tg(CMV-IL3,CSF2,KITLG)1Eav/MloySzJ (NSG-SGM3) Mice (JAX stock #013062) and NOD.Cg-*Prkdc^scid^ Il2rg^tm1Wjl^*/SzJ (NSG) Mice (JAX stock# 005557) were procured from Jackson Laboratory. The human IL6-expressing NSG (IL6) strain was generated in Dr. Shultz’s lab by microinjection of a transgenic construct containing the human interleukin-6, *IL6*, from the 161 Kb BAC clone RP11-469J8. This was microinjected into the pronuclei of fertilized NOD.CB17-*Prkdc^scid^*/J eggs. Founder line 1 was established and mated to NOD.Cg-*Prkdc^scid^ Il2rg^tm1Wjl^*/J (Stock No. **005557**) mice, and progeny were interbred to produce the NSG-IL6 strain.

The NSG-IL6/SGM3 mice were created by intercrossing the NSG-SGM3 and NSG-IL6 strains and using the F1 progeny that were hemizygous for the four human transgenes. We confirmed the genotype by PCR using the following primer sets (IL6, 5′ AGG GAG AGC CAG AAC ACA GA 3′ and 5′ TGC AGC TTA GGT CGT CAT TG 3′; CSF2, 5′ ACC TGC CTA CAG ACC CG 3′ and 5′ AGT GCT GCT TGT AGT GGC 3′; IL3, 5′ AAT CTC CTG CCA TGT CTG C 3′ and 5′ CCA GTC ACC GTC CTT GAT ATG 3′; KITLG, 5′ CAA GGA CTT TGT AGT GGC ATC TG 3′ and 5′ CAA CAG GGG GTA ACA TAA ATG G 3′).

### 2.3. Cell Culture and Maintenance

The U2932 and VAL DLBCL cell lines were maintained in RPMI 1640 medium supplemented with 15% fetal bovine serum (FBS), 1× penicillin/streptomycin, and 1× Glutmax (Gibco, Life Technologies, Carlsbad, CA, USA) at 37 °C in an atmosphere of 5% CO_2_. U2932 and VAL cells were cultured at a range of 0.5 to 2 × 10^6^ cells/mL. Cell lines purchased from DSMZ (U2932, ACC-633, RRID:CVCL_1896) or previously obtained from Dr. Rimsza, Mayo Clinic in Arizona, Scottsdale (VAL, RRID:CVCL_1819), and tested for mycoplasma every 6 months with the MycoAlert^®^ Mycoplasma Detection Assay (Promega, Madison, WI, USA) and authenticated by the University of Arizona Genetics Core (Tucson, AZ, USA) using the PowerPlex^®^ 16 System (Promega, Tokyo, Japan) every 10–12 months (Appendix A). Before injection into mice, cells were washed with PBS three times and counted by Cellometer Mini (Nexcelom, Lawrence, MA, USA) using the trypan blue exclusion method and transfected with a Luciferase lentiviral expression system.

### 2.4. Lentiviral Infection with the Luciferase Gene

To express Luciferase (Luc) in DLBCL cell lines, U2932 and VAL cells (1 × 10^5^ cells at >90% viability) were transduced with the lentiviral vectors encoding RedFLuc-T2A-EGFP or CBG-T2A-GFP carrying the firefly or beetle green *Luc* gene using TransDux MAX Lentivirus Transduction Enhancer (System Biosciences, Palo Alto, CA, USA) according to the manufacturer’s protocol. The GFP-positive cells were sorted after 72 h post-transduction by fluorescence-activated cell sorting (FACS) using a Becton Dickinson (BD) FACS Aria III-Cell Sorter (BD, San Jose, CA, USA).

### 2.5. DLBCL Cell Tail Vein Injection and In Vivo Bioluminescence Imaging

U2932-*Luc* and VAL-*Luc* cells (0.5 × 10^6^ cells at >90% viability) in 100 µL PBS were injected via the tail vein into NSG, NSG-IL6, NSG-SGM3, or NSG-IL6/SGM3 male and female mice at 8–12 weeks of age. An equal number of mice per sex was injected for each mouse strain. In vivo bioluminescence imaging (BLI) was conducted on a cryogenically cooled IVIS Imaging System 200 Series (Perkin Elmer, Waltham, MA, USA) using living image acquisition. Prior to BLI imaging, mice were anesthetized using isoflurane and imaged after 10 min of D-Luciferin (1.5 mg/mouse, Perkin Elmer, Waltham, MA, USA) via intraperitoneal injection. All the images were acquired by auto-exposure in the chamber under dim illumination, followed by acquisition and overlay of the pseudo color image representing the spatial distribution of photon counts produced by active Luc within the animal. Acquired images were analyzed using Living Image Software version 4.7.4 (Perkin Elmer, Shelton, CT, USA), region of interest (ROI) was generated to cover the whole body, and the total flux (*p*/s) was obtained. Although the second-week bioluminescence imaging confirmed DLBCL cell engraftment, the mice were continuously imaged once a week. The mice with no detectable signal by the fourth week post-injection were scored as ‘ungrafted’. All other mice were monitored for survival; the median survival of the mice strains was estimated by plotting Kaplan–Meier curves, and the Log-rank test was used to evaluate significant differences between control (PBS) and DLBCL cell-injected mice.

### 2.6. Ex Vivo Imaging

To evaluate the organ-specific engraftment/progression, we confirmed engraftment in the targeted organs by bioluminescence imaging starting in the second-week post-injection. From week 2 up to week 6, 1~4 U2932-*Luc*- or VAL-*Luc*- or PBS-injected NSG-IL6 mice were euthanized each week. At 10 min after the Luciferin injection, the mice were euthanized, then the intestines, kidney, heart, stomach, spleen, liver, lungs, ovaries, brain, and spine were excised and underwent ex vivo IVIS imaging. Acquired images were analyzed using Living Image Software version 4.7.4, and region of interest (ROI) was generated to cover the organ area.

### 2.7. Immunohistochemistry

At weeks 2, 3, and 4 post-injection, representative areas of the spleen, liver, and lung of each mouse from the ex vivo imaging were fixed in 10% buffered formalin and embedded in paraffin using a Leica tissue processor. Sections were cut at 3 μm from the paraffin blocks and mounted onto Superfrost^®^ Ultra Plus slides for immunohistochemical staining for human CD20 and counterstained with hematoxylin. Tissues were deparaffinized with xylene and hydrated in a graded ethanol series to distilled water. The antigen retrieval step was performed by microwave treatment (5 min) in Tris–EDTA buffer (pH 9.0). Peroxidase Block Solution and Protein Block Solution were used sequentially to block endogenous peroxidase and to prevent unspecific labeling, respectively. Tissue sections were incubated 30 min at room temperature with a polyclonal rabbit anti-human CD20 (PIPA532313, Fisher Scientific, Waltham, MA, USA) at a 1:200 dilution. CD20 staining was visualized with diaminobenzidine (DAB) and hydrogen peroxide as substrate. Nuclear background staining was performed with Gill’s hematoxylin for 30 s. Brightfield images of the CD20 stained tissues were taken on a Zeiss AXIO Imager M2 microscope (Zeiss, Nashville, TN, USA) at 200× magnification.

### 2.8. Western Blot Analysis

U2932 and VAL cells were treated in vitro with human IL6 (78148, StemCell Technology, Vancouver, BC, Canada) at 50 ng/mL for 30 min and 60 min. Cells were harvested, washed with cold PBS, and lysed on ice with RIPA buffer (Boston BioProducts, Milford, MA, USA) and 1× HALT protease/phosphatase inhibitor (ThermoFisher, Waltham, MA, USA) for 30 min. Lysates were centrifuged at 14,000 rpm for 10 min at 4 °C. Protein concentrations were quantified using BCA assay (Pierce, ID, USA). For each lysate, 100 μg total protein was separated by SDS-PAGE (4–20% Mini-PROTEAN Precast Protein Gels, BIO-RAD) and transferred onto PVDF membranes. Membranes were probed with antibodies against TBP (ab818, Abcam; 1:2500), STAT3 (79D7, Cell Signaling Technology, Danvers, MA, USA; 1:2000), and p-STAT3 (Tyr705, Cell Signaling Technology; 1:1000). Secondary antibodies used were goat anti-rabbit Dylight 800 (PISA535571, Thermofisher) and goat anti-mouse Dylight 650, (PISA510174, Thermofisher). Blots were imaged using the Biorad ChemiDoc MP (BIO-RAD, Hercules, CA, USA), and images were analyzed using Image Lab software v6.1 (BIO-RAD).

### 2.9. Cell Proliferation Assay

U2932 and VAL cells were plated in 6-well plates at a cell density of 2.5 × 10^5^/mL and equilibrated overnight. Cells were treated with human IL6 at 50 ng/mL for 2 to 48 h. Viable cells were counted at each time point using trypan blue exclusion on a DeNovix CellDrop BF (DeNovix, Wilmington, DE, USA).

### 2.10. qPCR Analysis

Untreated U2932 and VAL cells were subjected to real-time quantitative PCR (qPCR) analysis to measure mRNA abundance of IL6 receptor subunit α (CD126; Taqman, Hs01075664_m1) and gp130 (Taqman, Hs00174360_m1). Total RNA was isolated with a Roche High Pure RNA isolation kit (San Francisco, CA, USA), and reverse transcription was performed using the BioRad iScript kit according to the manufacturer’s protocols. qPCR was conducted with the BioRad Probe Supermix and TaqMan Probes using the BioRad CFX1000 Touch thermal cycler. The Ct values were normalized to TATA-binding protein (*TBP*; Taqman, Hs00427620_m1) and compared to the untreated controls to obtain ∆Ct values.

### 2.11. Statistical Analyses

For statistical analyses and plots, Student’s *t*-test, log-rank test, or one-way ANOVA were performed using Prism 7 or 9 (GraphPad Software Inc., San Diego, CA, USA) and Sigma Plot v13.0 (Systat Software Inc., San Jose, CA, USA).

## 3. Results

### 3.1. Luc-Expressing DLBCL Cells Engraft and Expand Rapidly in an Intravenous Xenograft NSG-IL6 Mouse Model

Since DLBCL is a genetically heterogeneous disease with at least two major molecular subtypes, we selected the well-established and characterized U2932 and VAL cell lines as representative of aggressive ABC- and GCB-DLBCL, respectively, to establish our in vivo model [18]. U2932 cells harbor extensive BCL2 amplification and mutated CD79B, while VAL cells are positive for BCL2, BCL6, and MYC translocations and CREBBP mutations, all of which are associated with poor patient outcomes [18,19,20,21,22,23,24,25]. To effectively monitor the engraftment and expansion of DLBCL in mice, we transduced the Luc-EGFP gene into DLBCL cells. The percentages of GFP-positive cells, 47.5% (U2932) and 28.7% (VAL), were exhibited at 72 h post-transduction.

A successful engraftment of DLBCL in NOD/scid mice was previously showing that subcutaneous passage enhanced the engraftment and metastatic capacity of various DLBCL cell lines [26]. Furthermore, successful engraftments of DLBCL cell lines and primary cells were reported in the humanized strains, MISTRG and MISTRG6, that demonstrated the importance of human IL6 in the in vivo engraftment of a subset of DLBCL and the spleen infiltrations of U2932 cells occurred only at MISTRG engrafted with cord blood human hematopoietic stem cells or MISTRG6 [14,16]. The MISTRG and MISTRG6 strains were not commercially available. We chose the commercially available NSG-SGM3 strain, a previously reported human hematopoietic cell engraftable strain [27,28,29], the recently developed NSG-human IL6 strain, and the F1 strain (NSG-IL6/SGM3). A total of 0.5 × 10^6^ GFP-positive U2932 or VAL cells (>90% viability) were injected into NSG, NSG-IL6, NSG-SGM3, or NSG-IL6/SGM3 mice via the tail vein. In vivo BLI was performed weekly until mice developed endpoint criteria (i.e., body condition scoring <2 or significantly high bioluminescence (BL) signals on the whole body of >1010 radiance). We made the region of interest (ROI) in BLI to cover the whole body to measure the total flux (photons/sec, p/s). Mice with no detectable signal by the fourth-week post-injection were scored ‘ungrafted’. All other mice were monitored for tumor growth and survival.

BL signals in U2932-Luc cells were observed from the second-week post-injection in both NSG-IL6 and NSG-IL6/SGM3 mice (Figure 1A,B). As the DLBCL growth progressed, the BL signals increased, and the focal BL signal spread throughout the body. While the BL of VAL-Luc cells was also observed from the second-week post-injection in NSG and NSG-IL6 mice, rapid tumor progression was noticed in comparison to the U2932-Luc cells. Most mice (9/11 mice) exhibited a high tumor burden by 4th week (Figure 1C,D). Of note, in further contrast to the U2932-Luc cells, there was no detectable engraftment of VAL-Luc cells in NSG-IL6/SGM3 mice despite some proliferation of cells near the injection site in weeks 6 and 7 (Figure 1C,D). These data suggest that the NSG-IL6 mice are highly permissive to both ABC and GCB subtypes of DLBCL cell lines. Furthermore, no engraftment for either U2932 or VAL cells was found in the NSG-SGM3 mouse strain (Figure 1A,C). In U2932-Luc cell-injected NSG mice, BL signals were shown in some mice (15/30 mice), whereas tumor growth in VAL-*Luc*-injected NSG mice rapidly progressed, and all the mice died by the 3rd week (Figure 1A,C).

### 3.2. Recipients of DLBCL Cell Intravenous Xenografts Show Poor Survival Corresponding to Tumor Burden

The rapid disease progression of VAL-Luc NSG-IL6-xenografted mice relative to the U2932-Luc NSG-IL6-xenografted mice corresponded to a poorer survival (Figure 2A; *p* = 0.0001; median survival 24 days vs. 49 days). We also observed a 75% (15/20) engraftment rate for U2932-Luc cells in NSG-IL6 mice and 32% (6/19) in NSG-IL6/SGM3 mice compared to 82% (9/11) for VAL-Luc cell engraftment in NSG-IL6 mice and 0% (0/10) in NSG-IL6/SGM3 mice (Figure 2B,C). Of note, both DLBCL cell lines in NSG-IL6 mice demonstrated uniform progression to death in the late phase of the disease. All mice injected with U2932-Luc and VAL-Luc died in a 16- and 9-day window, respectively.

### 3.3. DLBCL Intravenous Xenografts in NSG-IL6 Infiltrate Various Organs in Disease Progression

Previously, the subcutaneous passaged WSU in NOD/-scid mice demonstrated lymph node infiltration [26]. U2932 and primary DLBCL cells in the MISTRG mice engrafted with cord blood human hematopoietic stem (HSC) cells or non-engrafted MISTRG6 showed spleen infiltration [14]. To assess organ-specific engraftment of DLBCL cells in NSG-IL6 mice, we euthanized four xenografted mice per week at weeks 2–6 post-injection and performed ex vivo imaging of the excised organs. U2932-Luc cells were initially found in the spleen, liver, and lungs starting in week 2 (Figure 3A,B, left panels). Subsequently, U2932 cells were found in the ovaries of female mice, which is reflective of the parental tumor [19], spine, and brain (Appendix A). VAL-Luc cells also showed a similar pattern but occurred earlier (Figure 3D,E, right panels). Both U2932-Luc and VAL-Luc cells in NSG-IL6 mice exhibited a highly uniform tissue tropism that was unique to each cell line (spleen > liver > lung > brain > ovary). Next, we confirmed U2932-Luc and VAL-Luc infiltrations into the initially targeted organs and spleen, liver, and lung tissues by immunohistochemistry (IHC) staining for the human CD20 B cell surface marker. IHC staining showed a low CD20 positive cell infiltrate at the second-week post-injection that steadily increased at the third- and fourth-week post-injection (Figure 3C,F), consistent with the BL imaging (Figure 1 and Figure 3A,D). Notably, the NSG-IL6 mouse spleens at week 4 were dominated by the U2932-Luc cells. These findings confirm the early and highly permissive IL6-expressing mice for the engraftment and growth of DLBCL cells in organs consistent with clinical disease progression.

### 3.4. IL6 Induces Phosphorylation of STAT3 in DLBCL Cells

To confirm evidence that IL6 signaling is a critical driver of a subset of DLBCL and an indicator of engraftment and growth in NSG-IL6 mice [19], we treated U2932 and VAL cells with human IL6 and determined STAT3 activation. We observed STAT3 activation in U2932 cells after 1 h incubation with human IL6, in agreement with the previous report, and further showed that phosphorylation of STAT3 (pSTAT3) is inducible at a higher abundance after 30 min (Figure 4A). Although VAL cells show the constitutive activation of STAT3 as indicated by the presence of pSTAT3 in untreated cells, IL6 promotes additional STAT3 activation by ~6-fold after 30 min and to a lesser extent after 1 h. Previous work linked the expression of the IL6 receptor (IL6R) α subunit (IL6Rα) and gp130, the signaling chain subunit of the IL6R, to IL6-mediated STAT3 activation in DLBCL [14]. We confirmed that both IL6Rα and gp130 are expressed in U2932 and VAL cells, and we interestingly found VAL cells to express a significantly higher abundance of both subunits (Appendix A).

While human IL6 did not significantly increase U2932 cell proliferation, human IL6 increased VAL cell proliferation (Figure 4B), supporting the enhanced growth of VAL-Luc cells engrafted in NSG-IL6 mice compared to U2932-Luc cells. However, the basal proliferation of VAL cells compared to U2932 cells was significantly higher (*p* = 0.01, area under the curve), most likely owing to the aggressive proliferative phenotype associated with the so-called double- or triple-hit (BCL2/BCL6/MYC translocation, [25,30,31]) positive VAL cells, and thus, it also contributes to the more rapid expansion in NSG-IL6 mice.

## 4. Discussion

We developed a humanized mouse model that supports DLBCL engraftment and expansion in secondary lymphoid organs and extranodal sites (e.g., liver and lung) consistent with clinical presentation and further confirmed the pivotal role of human IL6 in human DLBCL cell homing in mice. A previous study demonstrated that ABC DLBCL cell lines engrafted and colonized the bone marrow and spleen only when MISTRG was engrafted with human HSCs or human IL6 was expressed in MISTRG (MISTRG6) [14]. These data suggest that human IL6 provides a more fertile environment for DLBCL cells, particularly subsets with inherent responsiveness to IL6 signaling, to engraft and expand in an in vivo model. However, it raises another question whether human IL6 alone is sufficient for the homing of human DLBCL cells in immune-deficient mice. Here, we evaluated DLBCL colonization and expansion in NSG mice expressing human IL3, CSF2, and KITLG with and without human IL6 co-expression and also in IL6-only-expressing NSG mice.

Our in vivo studies show that human IL6-alone-expressing NSG mice are permissive to the engraftment and organ colonization of the U2932 ABC and VAL GCB DLBCL cells, strongly indicating that IL6 is sufficient for systemic DLBCL tumor growth. Unlike MISTRG, a similar mouse, NSG-SGM3 mice, exhibited no engraftment of either ABC or GCB cells, whereas in the case of the NSG-alone engraftment of ABC cells was poor (50%), while GCB cell engraftment was rapid and vigorous. Such discrepancy may be derived from differences of other human cytokines or mouse backgrounds. However, without engraftment success rates in MISTRG or MISTRG6, we could not explain such a difference. Of interest, the NSG-IL6/SGM3 mice were somewhat permissive to U2932 engraftment (32%) and steadily increased tumor burden, leading to worsening survival.

Although traditional subcutaneous xenograft DLBCL mouse models allow for adequate drug discovery studies, the major limitation is the localized tumor growth in an often clinically non-relevant tissue that constrains the translation of how effectively new therapeutic strategies control tumor burden in humans. An in vivo model that more closely recapitulates the clinical progression of DLBCL within tissues that parallel the human disease provides a substantial advantage over these models. While spontaneous mouse lymphoma models overcome this limitation, these mouse tumors typically take months to develop and are driven by the enforced expression of specific oncogenes that fail to represent the molecular heterogeneity of DLBCL between subtypes and within a given tumor. Our approach of an intravenous injection of Luciferase-expressing human DLBCL cells in human IL6 transgenic NSG mice demonstrated rapid tumor cell engraftment and expansion, showing unique organ infiltration patterns reflecting the DLBCL subtype. We observed DLBCL cell infiltration and growth as early as two weeks post-injection, significantly faster than the MISTRG6 model [14]. Furthermore, despite the previous survey of GCB DLBCL cell lines demonstrating this subtype displays little to no IL6R expression, we show the VAL GCB DLBCL cells express a high abundance of both IL6R subunits. This finding suggests that not all GCB DLBCL cells are deficient in IL6 signaling and perhaps supports yet another aggressive phenotype of the double- or triple-hit DLBCL subset.

## 5. Conclusions

In summary, we establish an intravenous xenograft model for DLBCL that shortens the time to tumor development (median survival of U2932-Luc and VAL-Luc in NSG-IL6 mice; 49 days and 24 days), shows highly uniform tumor progression, follows a clinical progression of the disease, and requires minimal genetic alterations in mice with high potential to advance the preclinical testing of new therapeutics.

## Figures and Tables

**Figure 1 cancers-16-03006-f001:**
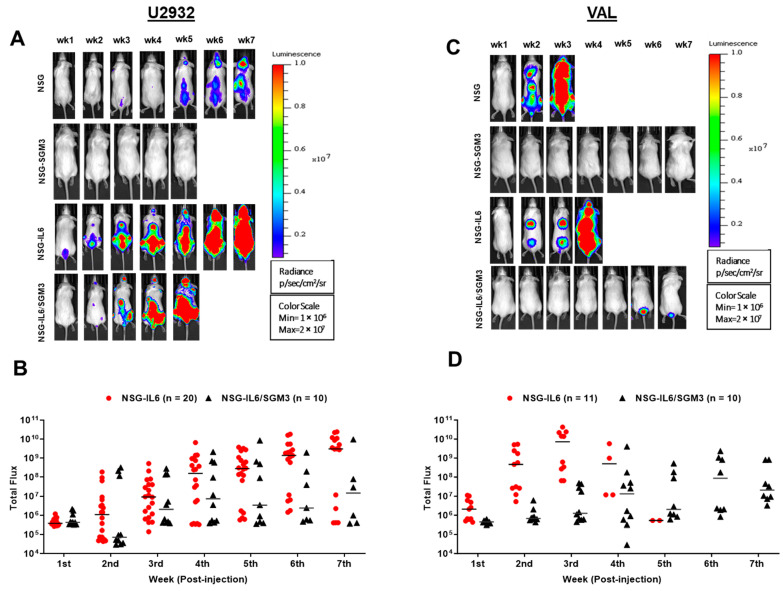
DLBCL cell line engraftment and tumor growth in NSG-IL6 and NSG-IL6/SGM3 mice. (**A**,**C**) Representative weekly BL images of (**A**) U2932-*Luc* and (**B**) VAL-*Luc* transplanted in NSG, SGM3, NSG-IL6, and IL6/SGM3 mice models. All signals were normalized in one scale as shown. A total of 5 × 10^6^ U2932-*Luc* and VAL-*Luc* cells in 100 µL of PBS were injected into the different strains of mice (8~12 weeks old) via the tail vein. (**B**,**D**) All total flux values (red circle or black triangle; NSG-IL6 or NSG-IL6/SGM3, respectively) with the median (line) are expressed as a progression of lymphoma. While no engraftment was observed in U2932- and VAL-transplanted NSG-SGM3 mice, NSG-IL6 mice for both and NSG-IL6/SGM3 mice of VAL showed engraftment.

**Figure 2 cancers-16-03006-f002:**
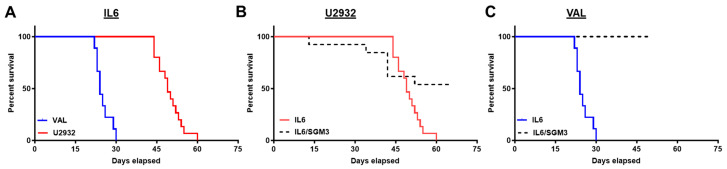
The survival comparison of intravenous DLBCL xenografts in NSG-IL6 mice. Mice were monitored weekly. When the mice met endpoint criteria, they were euthanized and survival recorded. (**A**) Survival curves representing 9 mice from VAL-*Luc*-engrafted NSG-IL6 mice (blue solid line) and 15 U2932-*Luc* NSG-IL6 mice (red solid line). The median survival was 24 days for VAL-*Luc*-engrafted NSG-IL6 mice compared to 49 days for U2932-*Luc*-engrafted NSG-IL6 mice. The Mantel–Cox test showed *p* = 0.0001. (**B**) The survival curve representing the comparison between U2932-Luc-engrafted NSG-IL6 and NSG-IL6/SGM3 mice. The median survival was 49 days for NSG-IL6 mice and undefined in NSG-IL6/SGM3 mice (*p* = 0.0294). (**C**) The median survival represents the NSG-IL6 mice vs NSG-IL-6/SGM3 mice in VAL-*Luc*-engrafted mice and recorded as 24 d vs. undefined (*p* = 0.0001).

**Figure 3 cancers-16-03006-f003:**
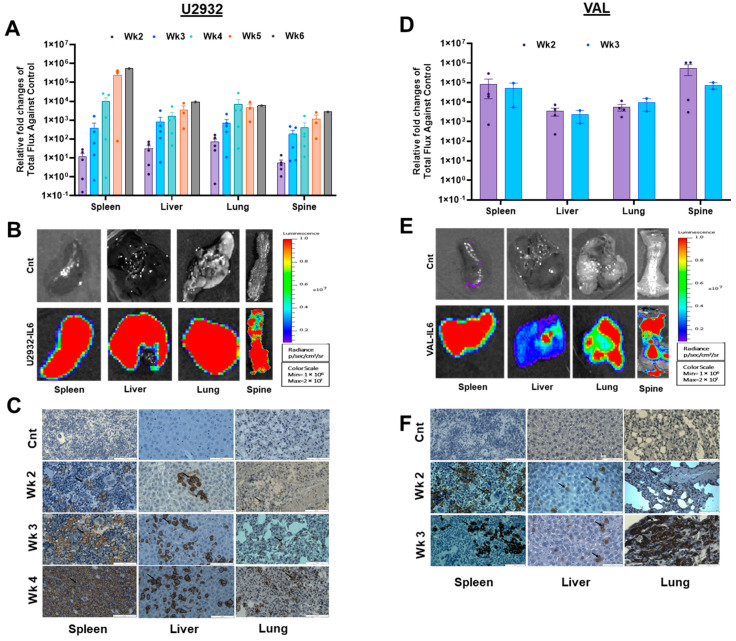
The ex vivo imaging and immunohistochemistry of DLBCL tumor growth according to the organ. (**A**,**D**) After DLBCL tumor cells were injected via the tail vein, we monitored engraftment/proliferation weekly using IVIS imaging. Representative (2~4) mice were selected weekly for ex vivo imaging. The selected mice were injected with D-Luciferin and euthanized after 10 min. Thereafter, the organs were excised and imaged with IVIS 200. All graphs show the ex vivo relative fold difference in organs regarding total flux against the control in U2932-*Luc*- and VAL-*Luc*-engrafted mice. The spleen, liver, lung, and spine showed the engraftment of U2932-*Luc* and VAL-*Luc* cells. (**B**,**E**) When whole-body BL signals reached above 5 × 10^9^ radiance (p/s), the mice were euthanized after 10 min of D-Luciferin injection. The organs were excised and imaged with IVIS 200. All images were normalized for BL signals in the scale shown here. (**C**,**F**) The expression of B cell lymphoma marker in U2932-*Luc*- and VAL-*Luc*-engrafted mice. The spleen, liver, and lung obtained from A and D were processed for immunohistochemistry. The expression was recorded by anti-CD20 antibody staining by immunohistochemistry. The arrow indicates the representative CD20 positive cells in the sections. The scale bar represents 50 µm.

**Figure 4 cancers-16-03006-f004:**
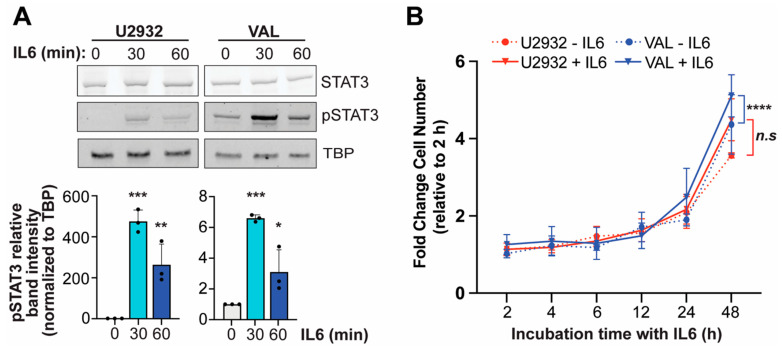
The IL6 activation of STAT3 and proliferation of DLBCL cells. (**A**) The Western blot analysis of STAT3 and phosphorylation of STAT3 (pSTAT3) in response to human IL6 treatment (50 ng/mL) for 30- or 60-min incubation in U2932 and VAL DLBCL cells. The quantification of pSTAT3 band intensity is shown below. Adjusted *p*-values as determined by a one-way ANOVA. * *p* < 0.05, ** *p* < 0.01, *** *p* < 0.001. The uncropped blots are shown in Appendix A. (**B**) The fold change in viable cell number after 2–48 h incubation with and without human IL6 (50 ng/mL) of U2932 and VAL DLBCL cells. Adjusted *p*-values as determined by a one-way ANOVA with Šídák’s multiple comparison test of the area under the curve (AUC). **** *p* < 0.0001. Data represents mean ± s.d. from three independent experiments.

## Data Availability

Protocols and data are available upon request.

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
