# Peer review of "Development of New Diffuse Large B Cell Lymphoma Mouse Models"

_cancers, 2024, doi:10.3390/cancers16173006_

Round 1
Reviewer 1 Report
Comments and Suggestions for Authors
This manuscript describes the characteristics of a new, humanized mouse-model of a frequent lymphoma. The study is well organized and includes genomic testing of transformed cells. I recommend its publication. However, as with any new model of a human disease the success rate of “grafted” neoplasm should be clearly stated. From the text I think that it is about 50% (P6,p2,L255-256) if so, it should be clearly emphasized in the abstract; also, the time-lapse of tumor development as well as mean time of disease progression (P10,L386-388) should be stated so that potential therapeutic studies can be conducted. The line “called lymphocyte” is unnecessary. (P1,L18).
Comments on the Quality of English LanguageNo comments.
Author Response
This manuscript describes the characteristics of a new, humanized mouse-model of a frequent lymphoma. The study is well organized and includes genomic testing of transformed cells. I recommend its publication. However, as with any new model of a human disease the success rate of “grafted” neoplasm should be clearly stated. From the text I think that it is about 50% (P6,p2,L255-256) if so, it should be clearly emphasized in the abstract;
Response to reviewer: added as suggested by the reviewer
also, the time-lapse of tumor development as well as mean time of disease progression (P10,L386-388) should be stated so that potential therapeutic studies can be
Response to reviewer: also added
Reviewer 2 Report
Comments and Suggestions for Authors
1- This is a well written and documented manuscript to understand DLBCL in more detail using mouse models. With the hope of providing better therapy.
2- Line 28, should be "Period" after "engraftment/progression.
3- The Figures are well constructed, and discussed, in particular Figures 1,3, and 4.
4- Supplementary material, particularly Scheme, provides more clarification.
5-
Author Response
- This is a well written and documented manuscript to understand DLBCL in more detail using mouse models. With the hope of providing better therapy.
Response: we appreciate valuable comments.
2- Line 28, should be "Period" after "engraftment/progression.
Response: Added
3- The Figures are well constructed, and discussed, in particular Figures 1,3, and 4.
Response: we appreciate valuable comments
4- Supplementary material, particularly Scheme, provides more clarification.
Response: we appreciate valuable comments
5- conducted. The line “called lymphocyte” is unnecessary. (P1,L18).
Response: removed